# The NGATHA-like Genes *DPA4* and *SOD7* Are Not Required for Stem Cell Specification during Embryo Development in *Arabidopsis thaliana*

**DOI:** 10.3390/ijms231912007

**Published:** 2022-10-09

**Authors:** Antoine Nicolas, Patrick Laufs

**Affiliations:** 1Université Paris-Saclay, INRAE, AgroParisTech, Institut Jean-Pierre Bourgin (IJPB), 78000 Versailles, France; 2Université Paris-Saclay, 91405 Orsay, France

**Keywords:** embryo, stem cells, meristem, CLV3, NGAL, DPA4/NGAL3, SOD7/NGAL2

## Abstract

In plants, stem cells are embedded in structures called meristems. Meristems can be formed either during embryogenesis or during the plant’s life such as, for instance, axillary meristems. While the regulation of the stem cell population in an established meristem is well described, how it is initiated in newly formed meristems is less well understood. Recently, two transcription factors of the NGATHA-like family, DEVELOPMENT-RELATED PcG TARGET IN THE APEX4 (DPA4)/NGAL3 and SUPPRESSOR OF DA1-1 7 (SOD7)/NGAL2 have been shown to facilitate de novo stem cell initiation in *Arabidopsis thaliana* axillary meristems. Here, we tested whether the *DPA4* and *SOD7* genes had a similar role during stem cell formation in embryo shoot apical meristems. Using *DPA4* and *SOD7* reporter lines, we characterized the expression pattern of these genes during embryo development, revealing only a partial overlap with the stem cell population. In addition, we showed that the expression of a stem cell reporter was not modified in *dpa4-2 sod7-2* double mutant embryos compared to the wild type. Together, these observations suggest that *DPA4* and *SOD7* are not required for stem cell specification during embryo shoot apical meristem initiation. This work stresses the difference in the regulatory network leading to meristem formation during the embryonic and post-embryonic phases.

## 1. Introduction

Meristems are the ultimate source of all plant organs and tissues, whether roots, stems, leaves or flowers [1,2]. These multicellular structures are either formed during embryogenesis—such as the shoot apical meristem (SAM) and the root apical meristem—or arise throughout the life of the plant, as is the case for lateral root meristems and axillary meristems. These newly formed meristems allow branching of the root and shoot system and are therefore essential to set up the plant’s architecture. Whatever their origin and their location, meristems have a conserved functional logic that enables both their maintenance and the production of new cells that will allow organogenesis and tissue formation. Central to this dual function is a population of stem cells which infrequently divide to replenish themselves and provide cells that will enter differentiation. Understanding how such a stem population is regulated and how it is formed is a central question in biology.

The SAM formed during embryogenesis and the axillary meristems (AMs) formed during the post-embryonic phase share a similar organization and regulation. These often dome-shaped structures contain at their summit a population of stem cells lying within a central zone (CZ) [3]. Division of CZ cells feeds peripheral cells which are involved in organ primordium formation. The population of stem cells is regulated through interactions with its neighborhood, in particular the underlying organizing center (OC). These interactions have been extensively characterized and set the basis of meristem functioning. The *WUSCHEL* (*WUS*) gene encoding for a HOMEOBOX-like transcription factor is expressed in the OC and contributes to maintaining stem cells in an undifferentiated and pluripotent state. WUS moves through the plasmodesmata to the CZ to induce the expression of the *CLAVATA 3* (*CLV3*) gene coding a small secreted peptide [4,5,6,7]. WUS ensures the formation of the stem cell niche according to its dimerization state [8,9,10,11]. In reply, CLV3 inhibits *WUS* expression upon binding to receptor kinases such as CLV1, CLV2, CORYNE (CRN) [12,13,14,15,16,17]. Others regulators, such as the HAIRY MERISTEM (HAM) and ARABIDOPSIS THALIANA MERISTEM LAYER 1 (ATML1) transcription factors, the miR171, as well as auxin and cytokinin (CK), contribute to the WUS/CLV3 regulatory network [18,19,20,21,22,23,24,25,26,27,28,29]. This regulatory network ensures the maintenance of the stem cell homeostasis, defines the stem cell and stem cell niche and permits meristem activity in response to various environmental constraints [30,31,32]. While the regulation of the stem cell population in an established meristem is well studied, how such a population arises in AM or in the embryonic SAM is far less understood.

Axillary meristems arise from the axillary region of leaves, which contains a few cells that remain meristematic while their neighbors differentiate into stem and leaf tissues. These cells retain their meristematic status through the action of transcription factors and particular hormonal signaling [33,34,35,36]. Central to this network are the meristematic markers, *SHOOT MERISTEMLESS* (*STM*) and the *CUP-SHAPED COTYLEDON 2* and *3* genes (*CUC2* and *3*), which are expressed in these cells and are required for AM initiation [36,37,38]. In parallel, auxin is actively depleted from these cells through polar auxin transport, which also contributes to maintaining their meristematic fate [34,39]. These meristematic cells remain latent in the axil until developmental and environmental signals trigger their proliferation [40]. A positive feedback loop between STM activity and cytokinin signaling has a major role in the increase of AM cell number. As the AM grows it becomes progressively organized into a typical shoot meristem. De novo expression of *WUS* is induced by CK and a permissive chromatin environment [41]. *CLV3* is induced by *WUS* in an initially overlapping domain. Later, *WUS* and *CLV3* expression undergo spatial rearrangement through HAM regulation to be eventually expressed in the OC and CZ, respectively [25,36,42]. However, very little is known about de novo initiation and establishment of stem cells in a developing meristem. Recently, it has been shown that the expression of *CUC2* and *CUC3* genes, which is initially observed in a strip of cells in the leaf axil, becomes reorganized into an eye-shaped pattern. Depletion of *CUC2* and *CUC3* expression from the centre of this eye-shaped domain is required for AM formation and de novo stem cell formation [42]. Furthermore, two NGATHA-like (NGAL) transcription factors DEVELOPMENT-RELATED PcG TARGET IN THE APEX4 (DPA4)/NGAL3 and SUPPRESSOR OF DA1-1 7 (SOD7)/NGAL2 have been shown to facilitate *CUC2* and *CUC3* repression in the emerging AM. Accordingly, *dpa4-2 sod7-2* double mutants show retarded AM formation, in particular delayed stem cell specification, a defect also observed in *dpa4-2 sod7-2* floral meristems, indicating a conserved role for these genes for all newly formed aerial meristems [42].

SAM initiation during embryo development shares many similarities with AM formation but differs by some of the actors involved. In particular, while *WUS* activates the formation of a stem cell population in AM, it does not play such a role in embryonic SAM initiation. Instead of *WUS*, a group of related factors defined as the *WOX2* module allows stem cell specification by repressing differentiation [43]. However, similar to that in the SAM, repression of the *CUC* genes in a central domain from which the embryonic SAM will emerge is required [44]. Since the *DPA4* and *SOD7* genes contribute to efficient AM formation and stem cells, this raises the question of whether these genes are also required for SAM formation in the embryo.

Here, we tested whether the *DPA4* and *SOD7* genes are required for stem cell specification in developing embryos. We also analyzed their expression pattern to test whether their expressions patterns were compatible with a role in meristem formation in the embryo. Altogether, our results indicate that in contrast to their role in AM formation, *DPA4* and *SOD7* are not required for stem cell formation in the embryo.

## 2. Results

### 2.1. Expression of a Stem Cell Reporter during Wild-Type and dpa4-2 sod7-2 Embryo Development

To determine whether stem cell formation in developing embryos required the activity of the *DPA4* and *SOD7* genes, we used the pCLV3:mCHERRY-NLS reporter to mark the stem cells and compared its expression in wild-type and *dpa4-2 sod7-2* double mutant embryos (Figure 1).

In the wild-type, pCLV3:mCHERRY-NLS expression was absent from embryos at the globular stage (Figure 1A,F). pCLV3:mCHERRY-NLS started to be expressed at the transition stage when the cotyledon primordia emerged (Figure 1B,C,G,H). Expression of pCLV3:mCHERRY-NLS was observed in a few cells in the middle of the embryo between the two emerging cotyledon primordia. At the heart stage, almost all embryos expressed pCLV3:mCHERRY-NLS (Figure 1D,I) and at the torpedo stage pCLV3:mCHERRY-NLS was detected in every embryo (Figure 1E,J). Our observations confirmed earlier reports that *CLV3* expression becomes detectable at the transition stage [43].

In the *dpa4-2 sod7-2* double mutant, pCLV3:mCHERRY-NLS showed an apparently similar expression timing and pattern as observed in the wild-type (Figure 1K–T). No pCLV3:mCHERRY-NLS was observed in embryos at the globular stage, while expression appeared at the transition stage (Figure 1K–T).

To quantify pCLV3:mCHERRY-NLS expression dynamics and reveal putative differences between the wild-type and the *dpa4-2 sod7-2* double mutant, we first counted the number of pCLV3:mCHERRY-NLS-expressing embryos at different stages (Table 1). This analysis did not reveal any significant difference in the frequency of pCLV3:mCHERRY-NLS-expressing embryos at different stages between the wild-type and *dpa4-2 sod7-2* double mutant. In particular, the proportion of pCLV3:mCHERRY-NLS-expressing embryos at the transition stage, the stage at which expression of the reporter became readable, was not different between the wild-type and the *dpa4-2 sod7-2* double mutant.

The results shown in Table 1 indicate that in the *dpa4-2 sod7-2* double mutant, as in the wild-type, stem cells become specified at the transition stage. To reveal possible differences in the timing of stem cell specification that could had been overlooked by the embryo categorization into developmental stages, we next analyzed pCLV3:mCHERRY-NLS activation regarding embryo size. For this, we measured embryo total length and then embryos were grouped into different bins based on their sizes and the frequency of pCLV3:mCHERRY-NLS expression in each bin was calculated (Table 2). Although we could observe some differences, notably a higher proportion of pCLV3:mCHERRY-NLS-expressing *dpa4-2 sod7-2* embryos in the bin containing the smallest embryo, none of these differences were statistically significant (Table 2).

To provide a more continuous analysis of pCLV3:mCHERRY-NLS activation dynamics, we plotted pCLV3:mCHERRY-NLS-expressing and non-expressing embryos in a morphospace based on embryo total length and embryo axis length (Figure 2A,B). In agreement with the results shown in Table 2, we observed that, for a certain size window (roughly between 40 and 90 µm total length), embryos with or without pCLV3:mCHERRY-NLS expression could be observed in both the wild-type and *dpa4-2 sod7-2*. Because of this variability, it was difficult to compare the dynamics of pCLV3:mCHERRY-NLS activation between the two genotypes. Therefore, we modelled pCLV3:mCHERRY-NLS activation (Figure 2C). For this, we first calculated the average frequency of pCLV3:mCHERRY-NLS-expressing embryos on a sliding window of 20 embryos. A fit of this average frequency showed no major difference between wild-type and *dpa4-2 sod7-2* (Figure 2C). In addition, when we calculated the embryo length at which half of the embryos had readable pCLV3:mCHERRY-NLS expression (L_50_), we obtained strikingly similar results for the wild-type and *dpa4-2 sod7-2* (54.6 and 54.8 µm, respectively).

Altogether, our analysis could not reveal any difference in the timing of pCLV3:mCHERRY-NLS activation during embryo development, whether staging the embryos on morphological features or on their size.

### 2.2. DPA4 and SOD7 Are Expressed during Embryo Development, but Show Different Dynamics of Expression

The absence of delay in stem cell specification in the *dpa4-2 sod7-2* double mutant raises the question of the expression of *DPA4* and *SOD7* genes during embryo development. *DPA4* expression has been described as initiating at the globular stage in a sagittal row of protodermal cells [43]. *SOD7* expression pattern during embryo development has not yet been described [45,46]. Therefore, we characterized *DPA4* and *SOD7* expression in developing embryos using pDPA4:GFP and pSOD7:GFP transcriptional reporters [42] (Figure 3 and Figure 4).

In the wild-type, pDPA4:GFP expression was visible at the globular stage in about half of the observed embryos (Figure 3A,B,F,G; Table 3). The expression of pDPA4:GFP was first observed in a single cell of the apical region of the embryo (Figure 3B,G) and later it expanded, forming a line of cells on the outer layer of the embryo (not shown). At the heart and torpedo stages pDPA4:GFP was detected in every embryo (Figure 3C–E,H,I,J). The longitudinal and the transverse views of the embryo showed that pDPA4:GFP was expressed in a few cells in the upper layer of the embryo and formed a strip of expression that delimits the two emerging cotyledon primordia (Figure 3C,D,H,I).

The pSOD7:GFP reporter showed a similar expression pattern, but with a different timing of expression compared to pDPA4:GFP. In contrast to the expression of pDPA4:GFP, expression of pSOD7:GFP was absent from all inspected embryos at the globular stage (Figure 4A,F, Table 3). pSOD7:GFP started to be expressed at the transition stage; however, only a small proportion of embryos expressed it (Figure 4B,C,G,H, Table 3). When it was expressed, pSOD7:GFP could be observed between the two emerging cotyledon primordia (Figure 4C,H). At the heart stage, again, less than half the embryos expressed pSOD7:GFP (Figure 4D,E,I,J; Table 3). Finally, at the torpedo stage pSOD7:GFP was detected in all embryos (Table 3).

Calculating the proportion of embryos expressing pDPA4:GFP or pSOD7:GFP in a given developmental stage (Table 3) or size class (Table 4) confirmed that pDPA4:GFP was expressed before pSOD7:GFP. As for the expression of *CLV3*, we plotted pDPA4:GFP/pSOD7:GFP expressing or non-expressing embryos in a morphospace (Figure 5A,B) and modelled frequency of reporter expression (Figure 5C). This clearly showed the difference in the timing of pDPA4:GFP or pSOD7:GFP activation illustrated by L_50_ of 37.6 µm for pDPA4:GFP and 68 µm pSOD7:GFP (Figure 5C).

*DPA4* and *SOD7* belong to the *NGAL* gene family that contains three members in Arabidopsis [47,48]. The third member, ABS2, has been shown to have redundant roles with *DPA4* and *SOD7* during leaf development [49]. Therefore, we wondered whether *ABS2* might be redundant with *SOD7* and *DPA4* in the de novo establishment of stem cells in embryos. To test this, we analyzed *ABS2* expression during embryo development using published transcriptomic data obtained from laser-assisted microdissected embryos at different developmental stages [50]. These transcriptomic data showed that the mRNA levels of the third *NGAL* gene ABS2 remained low throughout the entire morphogenetic phase of embryo development and increased only during the maturation (Figure 6). On the other hand, *DPA4* and *SOD7* were expressed early on during embryo development. This suggests that *ABS2* is not involved in the establishment of stem cells that are set up before *ABS2* is expressed.

## 3. Discussion

The initiation and development of stem cells remains largely unknown. Recently, it has been shown that two *NGAL* genes, *DPA4* and *SOD7*, facilitate *CUC2* and *CUC3* repression in the developing AM, and that this repression is required for de novo stem cell formation [42]. Because the AM shares with the SAM a phase of *CUC* gene repression during the early stages of its formation [42,44], it seemed important to check for *NGAL* contribution during stem cell formation in embryos. Therefore, we quantified the expression dynamics of *CLV3* in wild-type and *dpa4-2 sod7-2* embryos, but could not reveal any difference. Furthermore, we show that while *DPA4* is expressed before *CLV3* activation, the expression of *SOD7* occurs often later. Altogether, our data suggest that *DPA4* and *SOD7* are not required for de novo stem cell establishment during embryo development.

The absence of delayed *CLV3* expression in *dpa4-2 sod7-2* could be due to a higher level of genetic redundancy. Indeed, three *NGAL* genes are present in the Arabidopsis genome [47]. *DPA4* and *SOD7* have been suggested to result from a recent duplication event [47]. Accordingly, these genes have redundant roles during seed development [45], phyllotaxis control [46], leaf development [49] and AM formation [42]. However, in contrast to such redundant roles, our data suggest these two genes already initiated divergent evolution as we observed differences in their expression timing during embryo development. *ABS2*, the third *NGAL* gene, could have a redundant role with *DPA4* and *SOD7* for the promotion of stem cell formation in embryos. Although we cannot exclude it as we have not analyzed stem cell formation in the triple *ngal* mutant, it is unlikely as *ABS2* is expressed late during embryo development, after the phase of stem cell specification.

The absence of difference in the timing of *CLV3* expression between wild-type and *dpa4-2 sod7-2* double mutant could be explained by a hypothesis other than redundancy between the three *NGAL* genes. The formation of a new meristem involves a transition phase, called premeristem, during which either the regulators, their interactions or their expression patterns are different from the ones observed in established meristems (for a review, see [51]). Indeed, in established SAM, *WUS* induces *CLV3* expression, while in the embryo, other genes of the *WUS* family, *WOX1*, *WOX2*, *WOX3* and *WOX5*, are necessary for the activation of *CLV3* expression (Mayer et al., 1998; Zhang et al., 2017). The *wox1 wox2 wox3 wox5* quadruple mutation has variable effects on developing seedlings, leading to delays or complete absence of apical meristem initiation. Conversely, *CLV3* is still expressed normally in the *wus* mutant (Zhang et al., 2017). Neither in the *dpa4-2 sod7-2* double mutant nor the *dpa4-2 sod7-2 abs2-1* triple mutant seedlings did we observe a phenotype similar to the *wox* quadruple mutant (not shown), suggesting that *NGAL* genes do not affect *WOX* expression. Altogether, this suggests that while *DPA4* and *SOD7* genes are required for proper expression of the *WUS* gene to form the stem cell population in developing AM, they do not have a similar role for the regulation of the expression of the *WOX* genes promoting stem cell formation in developing embryonic SAM. Thus, the role of *DPA4* and *SOD7* during the development of the embryo remains unknown and needs to be determined. Altogether, this work stresses difference in the regulatory networks leading to meristem formation during the embryonic and post-embryonic phases.

## 4. Materials and Methods

### 4.1. Plant Material

The pCLV3:mCHERRY-NLS reporter was generated by [31]. The pCLV3:mCHERRY-NLS *dpa4-2 sod7-2* lines and the pDPA4:GFP and pSOD7:GFP lines were described in [42]. Because *dpa4-2 sod7-2* shows reduced fertility, the pCLV3:mCHERRY-NLS *dpa4-2 sod7-2* line was hand pollinated to ensure homogeneous seed formation.

### 4.2. Embryo Dissection and Confocal Microscopy

Seeds at different stages were gently squashed with a coverslip in a buffer (Tris-HCl 10 mM pH = 8.5, Triton 0.01%) containing 20% of the antifadent agent Citifluor.

Imaging was performed on a Leica SP5 inverted microscope (Leica Microsystems, Wetzlar, Germany). Stacks of 512 pixel-wide pictures with a Z-step of 2 µm were observed using a 20× lens. The pCLV3:mCHERRY-NLS lines were imaged with an excitation light of 561 nm and fluorescence was collected in a 590–650 nm window. The pDPA4:GFP and pSOD7:GFP lines were imaged with an excitation light of 488 nm and fluorescence was collected in a 518–554 nm window. Transmission images were collected to have access to the embryo morphology.

### 4.3. Image Analysis

Images were analyzed using Fiji [52]. Z-axis maximal projections were created to illustrate representative expression patterns.

Embryo axis length was determined manually using Fiji as the distance from the root pole to the most apical region lying between the cotyledon primordia or as the diameter of globular embryos. Length of the cotyledon primordia was measured along an axis parallel to the root/hypocotyl axis. Total embryo length was calculated as the sum of the embryo axis length and the length of the cotyledon primordia. The globular stage was defined as round embryos with no visible cotyledon primordia. The transition stage was defined as embryos with visible cotyledon primordia of which the size was < 5 µm. The heart stage was defined as embryo with an axis (root pole and hypocotyl) < 80 µm. The torpedo stage was defined as embryos with an axis ≥ 80 µm. Expression of the different reporters was examined on the different sections of the Z-stack, using the Fire LUT for samples with low signal intensity.

### 4.4. Data Analysis

Data were analyzed using R [53]. Data were represented using the ggplot2 package [54]. Averages of reporter expression were calculated on a 20 embryo-wide sliding window using the rollmean function from the zoo package [55]. Calculation of the L_50_ values for which half the embryos expressed the reporter was performed accordingly to the script described in [42] and available at https://doi.org/10.57745/ZQZVPD (accessed on 1 September 2022).

### 4.5. Gene Expression Levels

Expression levels of *ABS2*, *DPA4* and *SOD7* in the embryo at different developmental stages were retrieved from the gene Chip data generated by [50] at http://seedgenenetwork.net/ (accessed on 7 September 2022). Belmonte et al. (2013) microdissected different regions from seeds at different developmental stages and determined mRNA accumulation in these samples by hybridization to an ATH1 DNA Chip [50]. Data from the “embryo proper” were selected here.

## Figures and Tables

**Figure 1 ijms-23-12007-f001:**
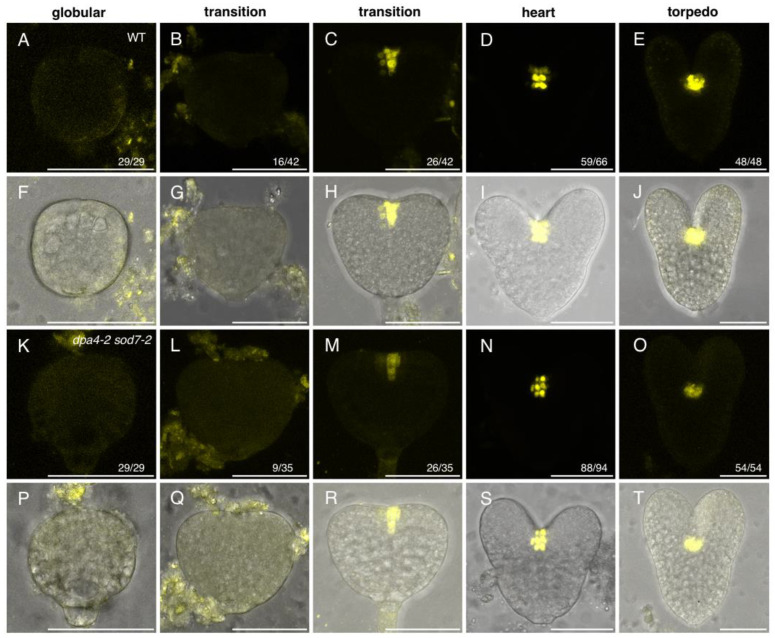
pCLV3:mCHERRY-NLS expression in developing wild-type and *dpa4-2 sod7-2* embryos. Panels (**A**–**J**) show wild-type embryos, while panels (**K**–**T**) show *dpa4-2 sod7-2* embryos. For each genotype, the upper row shows pCLV3:mCHERRY-NLS signal (in yellow), while the lower row shows an overlay of the pCLV3:mCHERRY-NLS signal (in yellow) and the transmission picture (in grey). Images are maximum projection form stacks of confocal images. The number of similar observations out of the total number of observations is shown in the lower right corner as a fraction. The embryo stage is indicated above the panels. For the transition stage, embryos without or with pCLV3:mCHERRY-NLS expression are shown. Scale bars = 50 µm.

**Figure 2 ijms-23-12007-f002:**
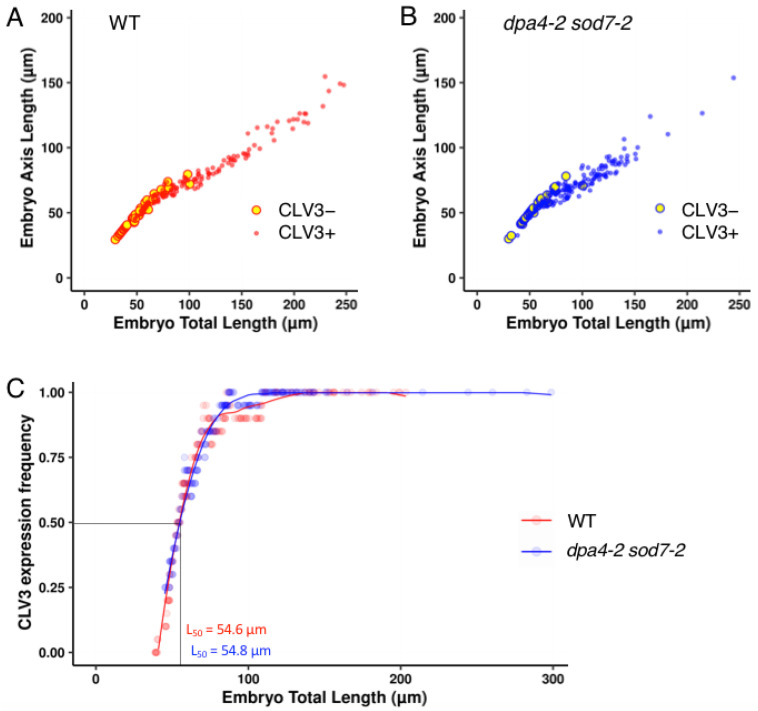
Dynamics of pCLV3:mCHERRY-NLS activation in developing wild-type and *dpa4-2 sod7-2* embryos. (**A**) Wild-type embryos not expressing pCLV3:mCHERRY-NLS (large yellow circles with red outline) or expressing it (small red dots) were plotted in a morphospace with embryo total length along the x-axis and embryo axis length along the y-axis. (**B**) *dpa4-2 sod7-2* embryos not expressing pCLV3:mCHERRY-NLS (large yellow circles with blue outline) or expressing it (small blue dots) were plotted in a morphospace with embryo total length along the x-axis and embryo axis length along the y-axis. (**C**) Model of pCLV3:mCHERRY-NLS activation during wild-type (red) or *dpa4-2 sod7-2* (blue) embryo development. The dots represent the average of the frequency of pCLV3:mCHERRY-NLS-expressing embryos calculated on a 20 embryo-wide sliding window. The red and blue lines represent curves fitted on these points. The total embryo length at which half of the embryo expressed pCLV3:mCHERRY-NLS (L_50_) was calculated (shown with dotted black lines).

**Figure 3 ijms-23-12007-f003:**
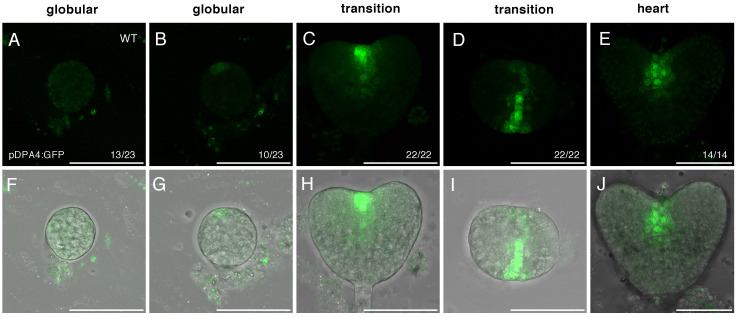
pDPA4:GFP expression in developing wild-type embryos. Panels (**A**–**E**) show the pDPA4:GFP signal (in green), while panels (**F**–**J**) show overlays of the pDPA4:GFP signal (in green) and the transmission picture (in grey). Images are maximum projection form stacks of confocal images. The number of similar observations out of the total number of observations is shown in the lower right corner as a fraction. The embryo stage is indicated above the panels. For the globular stage, an embryo without or with pDPA4:GFP expression are shown (**A**,**F** and **B**,**G**, respectively). For the transition stage a longitudinal (**C**,**H**) and a transverse (**D**,**I**) section is shown. Scale bars = 50 µm.

**Figure 4 ijms-23-12007-f004:**
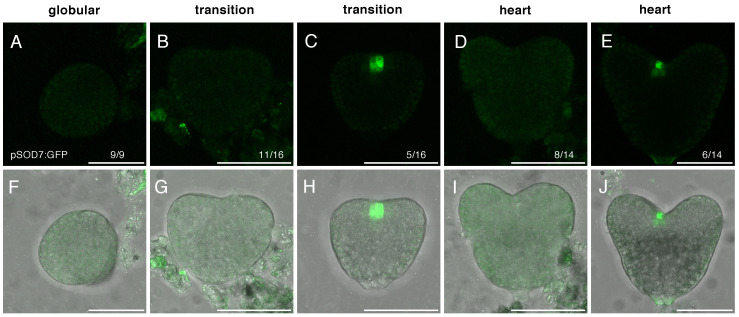
pSOD7:GFP expression in developing wild-type embryos. Panels (**A**–**E**) show the pSOD7:GFP signal (in green), while panels (**F**–**J**) show overlays of the pSOD7:GFP signal (in green) and the transmission picture (in grey). Images are maximum projection form stacks of confocal images. The number of similar observations out of the total number of observations is shown in the lower right corner as a fraction. The embryo stage is indicated above the panels. For the transition and the heart stages, an embryo without or with pSOD7:GFP expression are shown. Scale bars = 50 µm.

**Figure 5 ijms-23-12007-f005:**
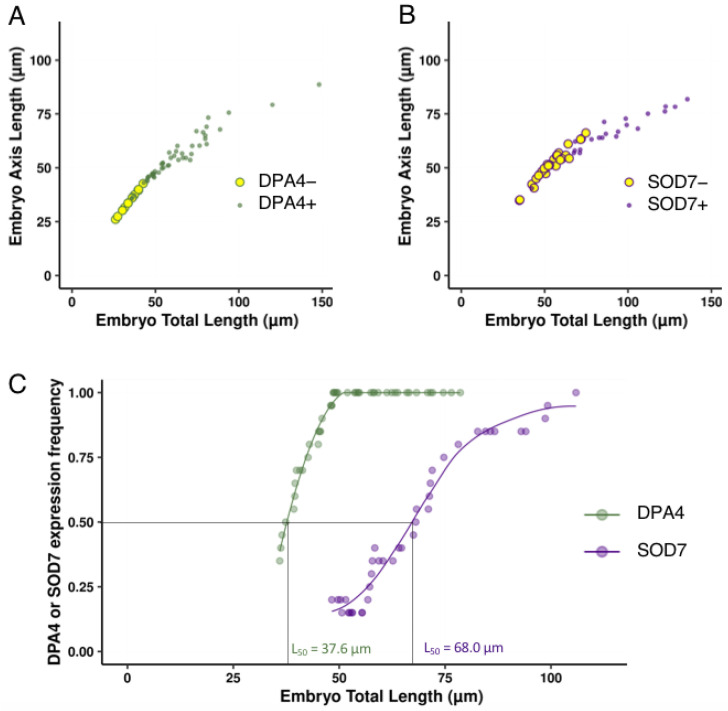
Dynamics of pDPA4:GFP and pSOD7:GFP activation in developing wild-type embryos. (**A**) Wild-type embryos not expressing pDPA4:GFP (large yellow circles with green outline) or expressing it (small green dots) were plotted in a morphospace with embryo total length along the x-axis and embryo axis length along the y-axis. (**B**) Wild-type embryos not expressing pSOD7:GFP (large yellow circles with purple outline) or expressing it (small purple dots) were plotted in a morphospace with embryo total length along the x-axis and embryo axis length along the y-axis. (**C**) Models of pDPA4:GFP (green) and pSOD7:GFP (purple) activation during wild-type. The dots represent the average of the frequency of reporter-expressing embryos calculated on a 20 embryo-wide sliding window. The green and purple lines represent curves fitted on these points. The total embryo length at which half of the embryo expressed pDPA4:GFP or pSOD7:GFP (L_50_) were calculated (shown with dotted black lines).

**Figure 6 ijms-23-12007-f006:**
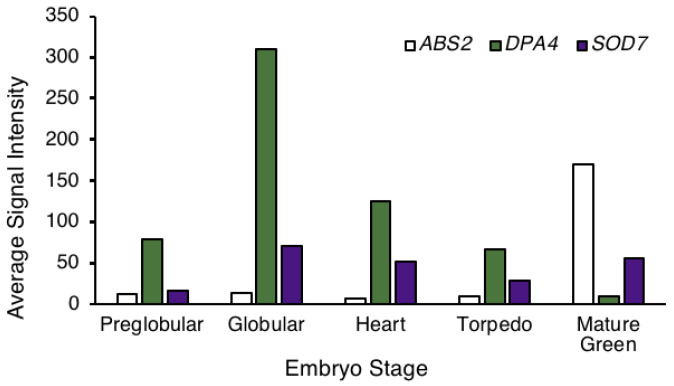
*ABS2*, *DPA4* and *SOD7* mRNA accumulation levels at different embryo developmental stages. Data were retrieved from [50].

**Table 1 ijms-23-12007-t001:** Frequency of *CLV3* expression in embryos depending on the developmental stage and the genotype.

Genotype	Stage							
	Globular		Transition		Heart		Torpedo	
WT	0%	(0/29)	62%	(26/42)	89%	(59/66)	100%	(48/48)
*dpa4-2 sod7-2*	0%	(0/29)	74%	(26/35)	94%	(88/94)	100%	(54/54)
*p*-value	-		0.3297		0.386		-	

The frequency of *CLV3* expression is shown as a percentage and the number of *CLV3*-expressing embryos compared to the number of observed samples is indicated in bracket. The *p*-value is the result of a Fisher’s exact test.

**Table 2 ijms-23-12007-t002:** Frequency of *CLV3* expression in embryos depending on the embryo size and the genotype.

Genotype	Embryo Total Length (µm)											
	]0–50]		]50–75]		]75–100]		]100–150]		]150–200]		]200–250]	
WT	9%	(3/32)	68%	(38/56)	88%	(29/33)	97%	(34/35)	100%	(18/18)	100%	(11/11)
*dpa4-2 sod7-2*	23%	(5/22)	63%	(43/68)	98%	(41/42)	98%	(57/58)	100%	(5/5)	100%	(2/2)
*p*-value	0.2479		0.705		0.1626		1		-		-	

The frequency of *CLV3* expression is shown as a percentage and the number of *CLV3*-expressing embryos compared to the number of observed samples is indicated in bracket. The *p*-value is the result of a Fisher’s exact test.

**Table 3 ijms-23-12007-t003:** Frequency of *DPA4* or *SOD7* expression in wild-type embryos depending on the developmental stage and the genotype.

Gene	Stage							
	Globular		Transition		Heart		Torpedo	
*DPA4*	43%	(10/23)	100%	(22/22)	100%	(14/14)	100%	(9/9)
*SOD7*	0%	(0/9)	31%	(5/16)	43%	(6/14)	100%	(19/19)

The frequency of *DPA4* or *SOD7* expression is shown as a percentage, and the number of *DPA4* or *SOD7*-expressing embryos compared to the number of observed samples is indicated in bracket.

**Table 4 ijms-23-12007-t004:** Frequency of *DPA4* or *SOD7* expression in wild-type embryos depending on the embryo size and the genotype.

Gene	Embryo Total Length (µm)									
	]0–50]		]50–75]		]75–100]		]100–150]		]150–200]	
*DPA4*	63%	(22/35)	100%	(21/21)	100%	(9/9)	100%	(2/2)	100%	(1/1)
*SOD7*	18%	(2/11)	30%	(8/27)	100%	(9/9)	100%	(4/4)	100%	(1/1)

The frequency of *DPA4* or *SOD7* expression is shown as a percentage, and the number of *DPA4* or *SOD7*-expressing embryos compared to the number of observed samples is indicated in bracket.

## Data Availability

Not applicable.

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
