# Peer review of "The NGATHA-like Genes DPA4 and SOD7 Are Not Required for Stem Cell Specification during Embryo Development in Arabidopsis thaliana"

_ijms, 2022, doi:10.3390/ijms231912007_

Round 1

Reviewer 1 Report

Using GFP reporter lines, Nicolas and Laufs characterized the expression pattern of two transcription factors (TF) belonging to the NGATHA-LIKE family, DEVELOPMENT-RELATED PcG TARGET IN THE APEX4 14 (DPA4)/NGAL3 and SUPPRESSOR OF DA1-1 7 (SOD7)/NGAL2, during zygotic embryo formation. These two genes have been recently shown by the same authors to facilitate de novo stem cell initiation in Arabidopsis thaliana axillary meristems. The authors showed that during embryo development the expression of these two genes only partially overlap with the stem cell population, and the expression of a stem cell reporter is not modified during dpa4 sod7 double mutant embryo development. These observations suggest the DPA4 and SOD7 are not required for stem cell specification during embryo shoot apical meristem initiation, differently from what occurs in vegetative axillary meristem formation.

These data are nicely presented and constitute an advance in understanding the genetic mechanisms of initiation and development of stem cells in plants.

The only part that is less convincing is the use of the in silico transcriptomic data from Laser-Assisted Microdissected embryos at different developmental stages to confirm the different timing of expression of DPA4 and SOD7. Although the expression of SOD7 appears lower than that of DPA4, the expression trends of the two TFs are quite similar at early stages of embryo development. Moreover, the main difference observed with the promoter-GFP fusions is at the globular stage, whereas both genes indeed seem to express from the transcriptomic dataset. I would therefore remove this part. However, if the Authors want to stress the difference between the expression of the three NGAL genes in the embryo, I would suggest to perform correlation analyses using all the expression values of the three genes during embryo development (embryo proper values) to stress their difference.

 Few minor revision of the text are needed.

Author Response

=> We agree with the reviewer that the dynamics of both genes are similar, but that the expression level of DPA4 is higher at early stages. Nevertheless, as it may be difficult to compare expression levels between genes, we have toned down the interpretation of the data coming from figure 6. Now, we only use these data to show that ABS2 is expressed to late to have a possible role during stem cell specification. We think that this is important and therefore would like to keep this figure.

Reviewer 2 Report

The study investigated the role of two genes, DPA4 and SOD7, in stem cell specification in developing embryos. Previous research showed that DPA4 and SOD7 inhibited CUC2 and CUC3 expression to facilitate axillary meristem (AM) formation and de novo stem cell formation. Could DPA4 and SOD7 genes have a similar role in de novo stem cell formation in the formation of the shoot apical meristem (SAM) in embryogenesis?

Substantive data showed that DPA4 and SOD7 were not required for initiating stem cells in the new SAM in embryogenesis. The double mutants dpa4-2 and sod-72 had no effect on the expression of the stem cell reporter CLV3 (pCLV3:mCherry-NLS signal) visualized during the development of the embryo SAM. Further,  pDBA-GFP and pSOD7-GFP showed different patterns of expression compared to the CLV3 stem cell reporter. DBA expression initiated earlier and showed a different expression pattern compared to SOD expression.

This study was well presented and the data clearly demonstrated that the N-GATHA-like genes DPA4 and SOD7 were not required for stem cell specification in embryogenesis. It was a useful investigation given the AM data previously obtained.

There are two points that should be commented on, and could be incorporated in the Discussion. First, what is the role of DPA4 and SOD7 given the time-course data obtained in embryogenesis. Second, in the case of the AM there is a complex vasculature. Given that the vasculature can house potential stem cells is this an important difference to the SAM initiation?

Author Response

=> The aim of this experiment was to correlate the expression of the NGAL genes with the expression of CLV3 during the different stages of embryo development to see if they were expressed at the same time and in the same areas. In the double mutant dpa4-2 sod7-2 we do not see any delay of CLV3 expression thus suggesting DPA4 and SOD7 do not regulate the expression of CLV3. Thus, their roles remain to be determined during embryo development. One possibility is that they regulate the expression of CUC genes, but this remains to be demonstrated.

=> As the reviewer suggests, stem cells are also present in the vascular system (cambium). Their regulation involves different actors as those operating in SAM and AM. We have not investigated whether the NGAL genes may be acting in this context, it is certainly something worth looking at but which extends the scope of this paper.